# The COMPASS Complex Regulates Fungal Development and Virulence through Histone Crosstalk in the Fungal Pathogen *Cryptococcus neoformans*

**DOI:** 10.3390/jof9060672

**Published:** 2023-06-14

**Authors:** Ruoyan Liu, Xiaoyu Chen, Fujie Zhao, Yixuan Jiang, Zhenguo Lu, Huining Ji, Yuanyuan Feng, Junqiang Li, Heng Zhang, Jianting Zheng, Jing Zhang, Youbao Zhao

**Affiliations:** 1College of Veterinary Medicine, Henan Agricultural University, Zhengzhou 450046, China; liuruoyan7588@163.com (R.L.); xiaoyu12052023@163.com (X.C.); zfj216403@163.com (F.Z.); jcc19990621@163.com (Y.J.); zhenguolu1999@163.com (Z.L.); lijunqiangcool@163.com (J.L.); zhrealm@163.com (H.Z.); 2State Key Laboratory of Microbial Metabolism, School of Life Sciences and Biotechnology, Shanghai Jiao Tong University, Shanghai 200240, China; myjsybysdp@163.com (H.J.); mn1590@126.com (Y.F.); jtzheng@sjtu.edu.cn (J.Z.); 3Joint International Research Laboratory of Metabolic and Developmental Sciences, Shanghai Jiao Tong University, Shanghai 200240, China

**Keywords:** COMPASS, H3K4 methylation, H2B ubiquitination, *Cryptococcus neoformans*, yeast-to-hypha transition, virulence

## Abstract

The Complex of Proteins Associated with Set1 (COMPASS) methylates lysine K4 on histone H3 (H3K4) and is conserved from yeast to humans. Its subunits and regulatory roles in the meningitis-causing fungal pathogen *Cryptococcus neoformans* remain unknown. Here we identified the core subunits of the COMPASS complex in *C. neoformans* and *C. deneoformans* and confirmed their conserved roles in H3K4 methylation. Through AlphaFold modeling, we found that Set1, Bre2, Swd1, and Swd3 form the catalytic core of the COMPASS complex and regulate the cryptococcal yeast-to-hypha transition, thermal tolerance, and virulence. The COMPASS complex-mediated histone H3K4 methylation requires H2B mono-ubiquitination by Rad6/Bre1 and the Paf1 complex in order to activate the expression of genes specific for the yeast-to-hypha transition in *C. deneoformans*. Taken together, our findings demonstrate that putative COMPASS subunits function as a unified complex, contributing to cryptococcal development and virulence.

## 1. Importance

The COMPASS complex regulates gene expression by mediating histone H3K4 methylation and has been confirmed to play a critical role in a variety of biological processes. Whereas the COMPASS complex has been studied in model yeasts, it has not been extensively explored in human fungal pathogens. Using a combination of AlphaFold modeling, molecular genetics, and biochemical approaches, we systematically identified the core subunits of the COMPASS complex in *C. neoformans* and *C. deneoformans*. The 3D modeling by AlphaFold-Multimer suggests that the core subunits form an integrated complex. Phenotypic and virulence assays reveal that the core subunits interact with each other and function in regulating cryptococcal morphogenesis and virulence traits. We further provide evidence that COMPASS-mediated H3K4 methylation activates transcript levels of genes that are related to the yeast-to-hypha transition in an H2B mono-ubiquitination (H2Bub1)-dependent manner. Therefore, this work demonstrates that the COMPASS complex epigenetically regulates cryptococcal development and virulence through histone crosstalk between H2Bub1 and H3K4 methylation.

## 2. Introduction

Methylation of histone H3 at lysine 4 (H3K4me) is an evolutionarily conserved post-translational modification that marks actively transcribed genes. H3K4 methylation, including mono-, di-, and tri-methylation (H3K4me1, H3K4me2, and H3K4me3), is catalyzed by the conserved Complex of Proteins Associated with Set1 (COMPASS) [1,2]. The COMPASS complex in *Saccharomyces cerevisiae* consists of seven distinct subunits, including Set1 (the methyltransferase subunit), Bre2/Cps60, Spp1/Cps40, Swd1/Cps50, Swd2/Cps35, Swd3/Cps30, Sdc1/Cps25, and Shg1/Cps15. Each subunit of the complex has a specific function, as some subunits contribute to the methylation activity while others regulate the assembly or stability of the complex [3,4,5,6]. Consistently, isolated Set1 alone from yeast is catalytically inactive and requires other subunits in the complex [7]. Swd1 and Swd3, two WD40 repeat-containing members of the complex, can stably associate with each other to form a heterodimer that is critical for maintaining global levels of H3K4 methylation [3]. Swd2, another WD40 repeat-containing protein, is essential in budding yeast [8,9] and is required for maintaining proper levels of H3K4me2 and H3K4me3 [7,10,11]. Bre2 and Spp1 are also required to achieve proper levels of H3K4me2 and H3K4me3 [12]. Bre2 directly interacts with Sdc1 [2,5,6], one of the smallest subunits of the complex. Spp1 and the n-SET domain (762–937) of Set1 are required for the stability of Set1 [10,12].

Many of these findings for the yeast COMPASS complex hold true for the human COMPASS complex. However, while Shg1 was identified as an integral subunit of yeast COMPASS, no mammalian homologs of Shg1 have been identified thus far, and the evolutionary loss of this subunit seems to have had little effect on COMPASS stability or functionality [2,13,14]. Given that different modules of the conserved COMPASS in yeast function collaboratively to mediate COMPASS assembly, cofactor binding, substrate recognition, and subsequent H3K4 methylation [1,7,15], it is surprising that the homolog of the functional subunit in yeast is dispensable in another organism. It is possible that to fulfill the multiple functionalities of different modules, this multi-component molecular machine might be intrinsically dynamic.

In fungi, accumulating evidence suggests that COMPASS-mediated H3K4 methylation regulates fungal development, stress responses, secondary metabolism, and virulence traits. In the plant fungal pathogen *Magnaporthe oryzae*, MoBre2, MoSpp1, and MoSwd2 function as a complex to methylate H3K4 in the region of the transcriptional start site of target genes that are involved in mycelial growth and the formation of conidia and infectious hyphae [16]. In *Alternaria alternata*, one of the most serious phytopathogenic fungi, AaSet1 also plays critical roles in cell development and is involved in the adaptation to cell wall interference agents and osmotic stress [17]. In *Fusarium graminearum*, FgSet1 is not only predominantly responsible for H3K4me, but it also plays an important role in response to cell wall-damaging agents via negatively regulating phosphorylation of FgMgv1, a core kinase in the cell wall integrity pathway [18]. FgSet1 has also been shown to associate components of the deoxynivalenol and aurofusarin biosynthesis pathways, indicating a regulatory role of COMPASS-mediated H3K4 methylation in secondary metabolism [18]. Deletion of COMPASS subunits in various *Aspergillus* species also leads to changes in the biosynthesis of several secondary metabolites [19,20,21,22]. In *Colletotrichum higginsianum*, the causal agent of crucifer anthracnose disease, H3K4me3 regulates pathogenicity and the production of three families of terpenoid secondary metabolites [23]. 

In the human fungal pathogen *Candida albicans*, disruption of *SET1* resulted in complete loss of H3K4me as well as hyper filamentous growth under embedded conditions and diminished adherence to epithelial cells; consequently, *SET1* was proposed to be associated with prolonged host survival and lower tissue fungal burdens [24]. In addition, Set1-mediated H3K4me is required for *C. albicans* to respond more rapidly to ROS generated by the host through activating mitochondrial protein-coding genes [25]. In another human fungal pathogen, *Candida glabrata*, Set1-mediated H3K4 methylation contributes to azole susceptibility by altering the expression of specific genes and consequently disrupting pathways known for mediating drug resistance [26]. Given the critical role of COMPASS-mediated H3K4me in regulating fungal development, stress responses, and pathogenesis, systematic identification of COMPASS subunits in other fungal pathogens would provide novel insights into regulatory mechanisms underlying morphogenesis and virulence and might inspire the development of novel infection control strategies.

*C. neoformans*, the top-ranked fungal pathogen in the Fungal Pathogen Priority List from the World Health Organization, is a globally distributed opportunistic fungal pathogen that is primarily of environmental origin. It is the major cause of meningitis in immunocompromised individuals, especially HIV-infected patients. Cryptococcal disease accounts for 19% of AIDS-related mortality, resulting in an estimated 112,000 cryptococcal-related deaths globally in 2020 [27]. *C. neoformans* is tolerant to human body temperature, facilitating its successful colonization in humans and other animal hosts [28,29]. Melanin production and capsule formation are two other virulence factors that allow the fungus to adapt to oxidative stresses generated by the host and to escape from the immune attack during infection [30,31,32]. In addition, the yeast and hyphal morphotypes of *C. neoformans* have been found to be linked to its pathogenicity [33].

Genetic regulatory pathways that are involved in cryptococcal morphogenesis and pathogenesis have been well-characterized; however, mechanisms of epigenetic regulation have been less well-studied. Here, we report the systematic identification of conserved subunits of the COMPASS complex in *C. neoformans* and *C. deneoformans*, including Set1, Bre2, Spp101 (a homolog of yeast Spp1), Swd1, Swd2, and Swd3, and we demonstrate that these conserved subunits interact with each other and function in mediating histone H3K4 methylation. Furthermore, our results show that these COMPASS subunits are involved in regulating the cryptococcal yeast-to-hypha transition, thermal tolerance, and melanin production, possibly through activating the transcription of genes specific for these biological processes. 

## 3. Materials and Methods

### 3.1. Strains, Growth Conditions, and Microscopic Examination

Strains used in this study are listed in Appendix A. *C. neoformans* and *C. deneoformans* strains were maintained on YPD medium unless specified otherwise. Transformants obtained from transient CRISPR-Cas9 coupled with electroporation (TRACE) were selected on YPD medium with 100 μg/mL of nourseothricin, 100 μg/mL of neomycin, or 200 μg/mL of hygromycin.

Strains for phenotypic assays were grown overnight in liquid YPD medium at 30 °C with shaking. The cells were washed with sterile water, adjusted to an optical density at 600 nm (OD_600_) of 3.0, and serially diluted. For filamentation tests, aliquots (3 μL) of cell suspensions (OD_600_ = 3.0) were spotted onto V8 plates and cultured at room temperature in the dark. For morphological examinations, all strains were examined under a stereoscope. For spotting assays, aliquots (3 μL) of serial dilutions starting from OD_600_ = 3.0 were spotted onto agar medium with supplements and cultured under the noted conditions.

### 3.2. Gene Manipulation

Cryptococcal genes were deleted following the TRACE protocol [34,35]. In brief, a deletion construct with approximately 1 kb of flanking sequences flanking a target gene and the split dominant marker was cloned through fusion PCR. This construct was mixed with PCR products of *CAS9* and a relevant guide RNA (gRNA), and the mixture was introduced into recipient strains by electroporation as described previously [35]. The resulting yeast colonies were screened by two rounds of diagnostic PCR. The first round of PCR was performed to detect the integration of the construct into the corresponding locus of the target gene. The second round of PCR was performed to confirm the knockout of the target fragment. All primers used to make gene deletion mutants are listed in Appendix A.

For gene complementation, the ORFs plus approximately 1.0 kb of their upstream regions were amplified by PCR and cloned into vectors through T5 exonuclease-dependent assembly as previously described [36]. For gene overexpression with inducible or constitutively active promoters, the constructs were created by amplifying the entire ORF by PCR and cloning the PCR products into vectors [37] behind the *CTR4*, *TEF1*, or *GPD1* promoter. All plasmids constructed were screened and confirmed by restriction enzyme digestion and sequencing. The confirmed constructs, together with PCR products of *CAS9* and gRNA targeting the Safe Haven locus [38,39], were introduced into recipient *Cryptococcus* strains. The transformants were passaged once per day for five days and cultured on selection plates to obtain stable transformants. Then, two rounds of diagnostic PCR were performed to confirm the integration and orientation of constructs into the Safe Haven locus. All primers and plasmids used for gene complementation and overexpression are listed in Appendix A.

### 3.3. Protein Domain Prediction and Phylogenetic Analysis

The InterPro option under the Protein Features menus in FungiDB [40] was used to predict the functional domains. For the phylogeny of COMPASS subunits, the whole-protein sequences were used to generate the neighbor-joining tree with MEGA7 [41]. The phylogeny for selected fungi was conducted with the Taxonomy Common Tree tool based on the classification of the NCBI Taxonomy database [42] and visualized with iTol [43].

### 3.4. Protein Extraction, Western Blotting, and Co-Immunoprecipitation Assay

Proteins were extracted from *Cryptococcus* cells according to a previously described method [44]. Aliquots of proteins were separated on 4%-to-12% gradient SDS-PAGE gels and then transferred to a polyvinylidene difluoride membrane for Western blot analyses. The protein bands from Western blot were relatively quantified with ImageJ by calculating the pixel density of bands. Briefly, the ratio of the pixel density of a protein band over the corresponding loading control band is calculated. This ratio from the control strain is set as 1, and ratios from mutants are normalized accordingly. Antibodies used in this study are listed in Appendix A. For co-immunoprecipitation assays coupled with mass spectrometry (CoIP/MS), mutants from *C. deneoformans* background were cultured overnight in liquid YPD at 30 °C with shaking. Whole-cell extracts of experimental strains were incubated with mNG-trap (ChromoTek) or FLAG-trap (Sigma, St. Louis, MO, USA) according to the manufacturer’s instructions. Proteins in the eluted samples were loaded onto an SDS-PAGE gel, digested, and analyzed by the proteome facility center of the Institute of Microbiology, Chinese Academy of Sciences. The protein-protein interactions were visualized with Cytoscape software following its standard manual [45].

### 3.5. 3D Modeling

The protein sequences of CnCOMPASS subunits from *C. deneoformans* were used to conduct 3D modeling. The structure of CnCOMPASS was predicted by Alphafold-Multimer (AF2-multimer) [46], which showed high accuracy in the multiple-chain protein structure prediction utilizing the deep learning algorithm. Then, this structure was submitted to the DeepUMQA [47,48] server to assess the quality. The global local Distance Difference Test (IDDT) [49] that measures the local distance differences of all atoms in the model of the predicted structure was 76.43.

### 3.6. RNA Extraction, RNA-Seq, and Quantitative Real-Time PCR Analyses

*Cryptococcus* strains were cultured in liquid YPD with shaking at 220 rpm at 30 °C overnight or on solid V8 medium at room temperature in the dark for 24 h. The cultures were collected, flash-frozen in liquid nitrogen, and lyophilized for 24 h. Total RNA was isolated with the PureLink RNA Mini Kit (Invitrogen, Waltham, MA, USA), and the first strand cDNA was synthesized using the GoScript Reverse Transcription System (Promega, Madison, WI, USA) following the manufacturer’s instructions. The Power SYBR Green system (Invitrogen) was used for RT-PCR. All the primers used here are listed in Appendix A. Relative transcript levels were determined using the ΔΔCt method with the *TEF1* gene as an internal control. Three biological replicates were included for all tests. Statistical significance was determined using a Student’s *t*-test, and *p*-values ≤ 0.05 were considered statistically significant.

### 3.7. Virulence Trait Assays

Strains for examining virulence factors were grown overnight in liquid YPD at 30 °C with shaking. The overnight cultures were washed with sterile water, adjusted to OD_600_ = 3.0, and serially diluted. For thermal tolerance and the melanin production formation assay, aliquots (3 μL) of serially diluted cell suspensions were spotted onto the YPD plate and L-dopamine media, respectively. Thermal tolerance was tested at 30 °C, 37 °C, and 39 °C. Melanin production was tested at 30 °C in the dark. Capsule formation was tested as described previously, and cells were strained with Indian ink for microscopic examination [50]. Titan cell formation was tested as described previously [50,51,52]. After inducing, cells were collected for microscopic examination, and cell size was determined with ImageJ. Data obtained were analyzed with the GraphPad Prism. All assays were repeated three times.

### 3.8. Galleria mellonella Larvae Model System

*Galleria mellonella* at the final instar larval stage were selected for infection as described previously [53]. *C. neoformans* strains were grown overnight in liquid YPD medium. Cells were washed with PBS and then resuspended in PBS to the final concentration of OD_600_ = 1.0. For infection, 5 μL of cell suspensions (~5 × 10^4^ cells) were injected into *G. mellonella* hemocoel through the last left proleg using a Hamilton syringe, and the control group was injected with 5 μL of sterile PBS. Before injection, the proleg was cleaned using an alcohol swab. Infected larvae were maintained in Petri dishes at 30 °C and monitored daily for survival by gently pressing down on pupae with the base of the pipette tip. Lack of movement following poking indicates death. Survival curves were generated with Kaplan–Meier analyses, and statistical analyses were conducted with Gehan–Breslow–Wilcoxon tests between the survival rate of each mutant and that of the H99 strain, as previously described [54].

For fungal burden analyses, infected larvae were maintained in Petri dishes at 30 °C for 5 days and then cleaned with 75% ethanol. The larvae were cut open with sterile scissors and vortexed in a microcentrifuge tube containing 500 μL PBS and 100 μL of 0.5 mm diameter glass beads. Larval suspensions were then serially diluted in PBS and plated onto YNB agar plates containing 100 μg/mL ampicillin. The plates were incubated at 30 °C for 2 days before counting the CFUs. All tests were repeated three times, and the statistical significance was determined with one-way ANOVA tests as previously described [54].

### 3.9. Murine Model of Cryptococcosis

Female BALB/c mice of 6–8 weeks old were purchased from the Laboratory Animal Center of Zhengzhou University, China. For fungal burden assays, 5 mice were assigned to each group. Cryptococcal strains were inoculated in 3 mL of YPD medium with the initial inoculum of approximately 10^6^ cells/mL. Cells were cultured overnight at 30 °C with shaking at 220 rpm. Cells were washed with sterile saline 3 times and adjusted to the final concentration of 2 × 10^6^ cells/mL. Mice were sedated with Ketamine and Xylazine via intraperitoneal injection and then inoculated intranasally with 50 mL fungal cell suspension (1 × 10^5^ cells per animal) as previously described [55,56,57]. After infection, animals were monitored daily for disease progression, including weight loss, gait changes, labored breathing, or fur ruffling. For lung fungal burden measurement, animals were euthanized on day 14 post-infection. The one-way ANOVA tests were used in the fungal burden studies as previously described [54].

### 3.10. Data Availability

All RNA-seq data are going to be available at the NCBI (SUB12944285).

## 4. Results

### 4.1. Identification of COMPASS Subunits in C. neoformans

Six potential subunits of COMPASS, including Set1, Bre2, Spp101, Swd1, Swd2, and Swd3, were identified in *C. deneoformans* based on amino acid sequence homologies to corresponding subunits of the yeast COMPASS complex. Functional domain prediction and phylogenetic analysis revealed that the six subunits of COMPASS in *C. deneoformans* contain functional domains conserved in their cross-species homologs (Figure 1A,B). It is notable that one of the yeast COMPASS subunits, Sdc1, was not identified in *C. deneoformans* through this initial sequence similarity analysis.

To further investigate the function of the COMPASS complex in *C. deneoformans*, we examined strains in which the genes coding for COMPASS subunits were deleted in the *C. deneoformans* XL280 reference strain background [58]. All deletions were confirmed by diagnostic PCR. To investigate the function of the COMPASS complex in mediating cryptococcal H3K4 methylation, we measured the H3K4 methylation by Western blotting assays. H3K4me1, H3K4me2, and H3K4me3 were almost completely abolished in *set1*Δ, *bre2*Δ, *swd1*Δ, and *swd3*Δ strains (Figure 1C), indicating that the corresponding proteins play critical roles in mediating H3K4 methylation. Deletion of *SPP101* dramatically impaired H3K4me1, H3K4me2, and H3K4me3, while deletion of *SWD2* slightly reduced the levels of H3K4me1, H3K4me2 or H3K4me3 (Figure 1C).

Next, we complemented the mutants with mNeonGreen (mNG)-tagged fusion proteins. In each case, the mNG-tagged constructs successfully complemented the H3K4 methylation deficiency exhibited by the corresponding deletion mutants (Figure 1C). Fluorescence analyses showed that the expression of each of the six subunits of COMPASS led to the enrichment of fluorescent signals in the nucleus (Figure 1D). The expression of mNG-tagged Swd1, Swd2, and Swd3 also produced more diffused fluorescent signals in the cytoplasm than the other three mNG-tagged subunits (Figure 1D).

### 4.2. Interaction between COMPASS Subunits in C. neoformans

In yeast, ScSet1, ScBre2, ScSwd1, ScSwd3, and ScSdc1 interact with each other and form a five-subunit catalytic core that is critical for the basal and regulated enzymatic activities of the COMPASS complex [5,6]. To reveal the architecture of the cryptococcal COMPASS catalytic module, we conducted 3D structural modeling based on the crystal structure of the intact yeast COMPASS catalytic module, although we could not identify a candidate *C. deneoformans* Sdc1 subunit based on amino acid sequence similarity. The 3D modeling results by AlphaFold-Multimer suggested that the core subunits of COMPASS may form a similar conformational architecture to function together as an integrated complex in *C. deneoformans* (Figure 2A). These findings suggested that the catalytic core of COMPASS in *C. deneoformans* may execute the conserved H3K4 methylation function, despite the apparent lack of Sdc1.

To confirm the interactions among cryptococcal COMPASS subunits and to search for the potential Sdc1 in *C. deneoformans*, we decided to identify their interacting protein partners. To that end, we overexpressed FLAG-tagged versions of Set1, Bre2, Swd1, and Swd3 in their corresponding deletion mutant strains to conduct a pull-down assay with anti-FLAG affinity beads. The results of phenotypic assays and Western blotting showed that the FLAG-tagged versions complemented the deficiencies in filamentation and histone methylation of the corresponding deletion mutations (Appendix A). 

We next performed a co-immunoprecipitation (CoIP) assay using anti-FLAG antibodies followed by mass spectrometry (MS) analyses to identify members of the relevant complexes. When Set1 was immunoprecipitated, all five of the other COMPASS subunits, including Bre2, Spp101, Swd1, Swd2, and Swd3, were co-immunoprecipitated with Set1 (Figure 2B, Appendix A), consistent with the formation of the catalytic core of COMPASS in *C. deneoformans*. Furthermore, reciprocal pull-down assays using FLAG-tagged Bre2, Swd1, and Swd3 identified the other putative complex subunits as interacting partners, and all three of these proteins co-immunoprecipitated Set1, Spp101, and Swd2 (Figure 2C, Appendix A), again supporting the formation of the catalytic core of COMPASS in *C. deneoformans*. 

A direct interaction between ScBre2 and ScSdc1 was previously identified in the crystal structure of the yeast COMPASS complex (Figure 2A). We hypothesized that a Sdc1 homolog in *C. deneoformans* would also interact with Bre2. Thus, we further investigated potential Bre2-interacting proteins from the FLAG-tagged Bre2 pull-down assay and prioritized them based on size, predicted function, and predicated sub-cellular localization. In total, we identified 41 potential CnSdc1 candidates and obtained mutants in *C. neoformans* H99 background from the Cryptococcus deletion set for 26 out of 41 and constructed deletion mutants in *C. deneoformans* XL280 background for the remaining 15 of them, as labeled in Appendix A. We then tested their impact on histone H3K4 methylation. However, none of the mutant strains exhibited deficiency in H3K4 methylation (Appendix A). 

We also conducted a BLAST search of genomes from Basidiomycota species and Ascomycota species in FungiDB using the amino acid sequence of ScSdc1 as a query. While we were able to identify homologs of ScSdc1 in most Ascomycota species, we were unable to find homologs of ScSdc1 in any of the Basidiomycota species cataloged in FungiDB (Figure 2D). These findings strongly suggest that the Sdc1 subunit might have been lost in the COMPASS complexes after the split of these two phyla in kingdom fungi. 

### 4.3. COMPASS Subunits Regulate Virulence Traits in C. neoformans

Thermal tolerance, melanin production, and capsule formation are the three classical factors that define the virulence of *C. neoformans*. To dissect the role of COMPASS in regulating cryptococcal virulence, we planned to investigate the effect of the deletion of COMPASS subunits on these cryptococcal virulence factors in the H99 reference strain background [63]. We obtained *set1*Δ, *bre2*Δ, *spp101*Δ, *swd1*Δ, and *swd2*Δ in the H99 background from the cryptococcal gene deletion set made in the H99 background by the laboratory of Hiten Madhani, and we deleted the *SWD3* gene in this background through TRACE. All deletions were confirmed by diagnostic PCR. As expected, deletion of the COMPASS components in the *C. neoformans* H99 background had similar effects on H3K4 methylation in this background as in the *C. deneoformans* XL280 strain background (Appendix A).

Most of the deletion mutants exhibited an apparent reduction in thermal tolerance, except that the *swd2*Δ strain exhibited a higher tolerance to 37 °C relative to other mutants (Figure 3A). The results of analyses of melanin in the mutants showed that almost all of the deletions led to decreased melanin accumulation; one deletion strain, *spp101*Δ, had a similar level of melanin production as *swd2*Δ (Figure 3B). 

Titan cell formation has been considered another factor that contributes to the virulence of cryptococcal infection [64,65,66]. To investigate the impact of COMPASS subunits on titan cell formation in *C. neoformans*, we determined the cell size of COMPASS mutants under three conditions that are known to induce titan cell formation [50,51,52]. The results showed that the formation of titan cells was normal in all mutants (Figure 3C, Appendix A), indicating that COMPASS subunits have no impact on titan cell formation in *C. neoformans*. The capsule production in both deletion and complementation strains was not visibly different in comparison to the wild-type H99 strain. The known capsule-overproducing *pas3*Δ strain [61] was used as a comparison (Figure 3D).

Furthermore, we used a *G. mellonella* model to test the effects of COMPASS subunit deletion on *C. neoformans* virulence in vivo. Here, *G. mellonella* larvae infected with COMPASS deletion mutants had prolonged survival times relative to the wild-type H99 control (Figure 3E). Consistently, all of the deletions of COMPASS subunits led to reduced fungal burdens on day 5 post-infection (Figure 3F). Finally, we confirmed the impact of COMPASS subunit deletion on cryptococcal virulence in the intranasal murine model of cryptococcosis. The results showed that the lung fungal burden of mice infected with COMPASS subunit deletion mutants 14 days post-infection is significantly lower than that of mice infected with the H99 reference strain (Figure 3G), consistent with what we observed in the *G. mellonella* model. Taken together, our results demonstrated that COMPASS subunits positively regulate thermal tolerance and melanin production in *C. neoformans* and also positively regulate virulence both in the *G. mellonella* model and the murine model of cryptococcal infection.

### 4.4. COMPASS Subunits Regulate Yeast-to-Hyphal Transition in C. deneoformans

*C. deneoformans* undergo yeast-to-hyphal transition in response to mating signals and environmental stimuli [62]. To investigate the role of COMPASS subunits in the cryptococcal yeast-to-hyphal transition, we cultured deletion mutants of COMPASS subunits from the self-filament *C. deneoformans* XL280 strain background to test the phenotype of yeast-to-hyphal transition. Interestingly, the deletion of COMPASS subunit coding genes dramatically impaired unisexual filamentation in *C. deneoformans* (Figure 4A). Introducing mNG-tagged versions of COMPASS subunits into the corresponding deletion mutants restored unisexual filamentation (Figure 4A). This result confirmed the role of these proteins in regulating the cryptococcal yeast-to-hypha transition. 

The transcription factor Znf2 is a master regulator that governs yeast-to-hypha transition in *C. deneoformans*. Its downstream target *CFL1* is considered a marker of cryptococcal filamentation [67]. To investigate the regulation of *ZNF2* and its regulon by COMPASS, we determined the transcript levels of *ZNF2* and its downstream targets *CFL1* and *FAD1* in COMPASS subunit mutants under unisexual filamentation-inducing conditions (on V8 media); wild-type XL280 cultured in filamentation-suppressing conditions (in liquid YPD) served as a control. As expected, the transcript levels of *ZNF2*, *CFL1*, and *FAD1* in the wild-type strain were significantly upregulated in cells grown on V8 media in comparison to those grown in liquid YPD (Figure 4B). Deletion of COMPASS subunits impaired the induction of *ZNF2*, *CFL1*, and *FAD1* at the transcript level, and overexpression of the genes encoding COMPASS subunits in the corresponding mutant strains restored their transcriptional induction (Figure 4B). Given the role of the COMPASS complex in positively regulating gene transcription through methylating H3K4, our results suggested that the COMPASS complex may facilitate the transcription of *ZNF2* through methylating H3K4 at specific loci of the genome to promote the cryptococcal yeast-to-hypha transition.

To further test the role of COMPASS in regulating bisexual mating, we obtained gene deletion mutants in the XL280 *MAT***a** [55] background through spore dissection following the crossing of *MAT*α mutant strains with the *MAT***a** wild-type strain. The mating type and target gene deletions were confirmed by diagnostic PCR (Appendix A). The results of bilateral mating assays (mutant α × mutant **a**) showed that bisexual filamentation was dramatically impaired in comparison to the mating control between wild-type strains (Figure 4C). 

The pheromone sensing pathway is critical for the success of bisexual mating in *C. deneoformans*. Thus, we used qPCR to quantify the transcript levels of the *MFα2*, *STE6*, and *STE3α*, which encode the α pheromone, the pheromone exporter, and the **a** pheromone receptor, respectively, in COMPASS subunit mutant strains. These assays clearly showed that deletion of COMPASS subunit-coding genes abolished the induction of these genes associated with the pheromone sensing pathway, although complementation of COMPASS subunits in the mutants restored the transcript levels of genes to varying degrees (Figure 4D). These results indicate that the COMPASS complex may promote cryptococcal bisexual mating by regulating the pheromone sensing pathway. Taken together, these results demonstrated that COMPASS subunits regulate both unisexual and bisexual filamentation in *C. deneoformans*.

### 4.5. COMPASS-Mediated H3K4 Methylation Requires H2Bub1 in C. deneoformans

Regulation of COMPASS-mediated H3K4 methylation by histone H2B mono-ubiquitination (H2Bub1) is an evolutionarily conserved trans-histone crosstalk mechanism [11,12]. The ubiquitin-conjugating enzyme Rad6 and the E3-ligase Bre1 covalently attach ubiquitin onto H2B K123 in a Paf1 complex (PAF1C)-dependent manner (Figure 5A) [60]. In previous studies, we found that H2Bub1 is abolished and H3K4me2 is significantly impaired in a *bre1*Δ mutant strain of *C. deneoformans* [61]. 

In the present study, to further investigate the crosstalk between H2Bub1 and H3K4 methylation in *C. deneoformans*, we created mutant strains lacking either *RAD6* or a gene (*RTF1*) that encodes a subunit of PAF1C. We subsequently found that deletion of either *RAD6* or *RTF1* abolished both H2Bub1 and H3K4me3; however, deletion of the COMPASS subunits *SET1* or *BRE2* only abolished H3K4 methylation and had no significant impact on H2Bub1 (Figure 5B), consistent with a model in which H2Bub1 is required for H3K4 methylation. Given the role of H3K4 methylation in regulating thermal tolerance and filamentation in *C. deneoformans*, *rad6*Δ and *rtf1*Δ showed dramatic deficiencies in thermal tolerance and filamentation (Figure 5C,D), suggesting their co-regulation on cryptococcal pathogenesis and morphogenesis. 

To further investigate the regulatory crosstalk between COMPASS-mediated H3K4 methylation and H2Bub1 on the cryptococcal yeast-to-hypha transition, we conducted RNA-seq analyses of *bre2*Δ, *rtf1*Δ, and *rad6*Δ mutant strains under filamentation-inducing and -suppressing conditions. The results showed that the three mutants shared more downregulated genes than upregulated genes (Appendix A), supporting their positive co-regulation on gene transcription through histone modification (Figure 6A,B). Given the positive regulation of H3K4 methylation and H2Bub1 on the cryptococcal yeast-to-hypha transition, we reasoned that genes that are involved in regulating this transition should be downregulated in all of the three mutants. Accordingly, the GO functional analyses of the shared downregulated genes showed that categories involving the promotion of the yeast-to-hypha transition, such as pheromone-dependent signal transduction, G-protein coupled receptor signaling pathways, and sequence-specific DNA binding transcription factors, were significantly changed (Figure 6C). As expected, *ZNF2*, *CFL1*, *STE6*, *STE3*, and other pheromone-associated genes were identified as downregulated genes in all three mutants (Figure 6A); reads coverage analyses of specific loci of these genes also clearly showed the downregulation of their expression (Figure 6D). In addition, *MAT2*, encoding the key regulator of pheromone sensing, was also significantly downregulated in *bre2*Δ, *rtf1*Δ, and *rad6*Δ mutant strains under filamentation-inducing (Appendix A), which was consistent with the downregulation of pheromone genes (Figure 6D). 

We also conducted RNA-seq with the *BRE2* complementation strain as a control, and the results of read coverage analyses showed the restoration of gene transcript levels (Figure 6D), further confirming that the COMPASS-mediated H3K4 methylation positively regulates the transcription of genes that govern the cryptococcal yeast-to-hypha transition. To validate the transcriptome results, we used qPCR to quantify the transcript levels of *ZNF2*, *CFL1*, and *MFα*, and the results were consistent with the RNA-seq results (Figure 6E). Taken together, all the lines of evidence support a model in which COMPASS-mediated H3K4 methylation regulates the yeast-to-hypha transition in an H2Bub1-dependent manner in *C. deneoformans*. 

## 5. Discussion

The Set1 family of histone methyltransferases regulates transcription by catalyzing the modification of H3K4me1, H3K4me2, and H3K4me3 through the formation of the conserved COMPASS complex. COMPASS-mediated H3K4 methylation is a key epigenetic event that plays important roles in a variety of cellular events, including gene expression, DNA replication and repair, chromatin compaction, and cell-cycle control. In higher eukaryotes, misregulation of H3K4 methylation has been implicated in the pathogenesis of cancer and in development defects [7]. In lower eukaryotes, H3K4 methylation has been shown to be required for morphogenesis, stress responses, secondary metabolism, and pathogenesis, particularly in pathogenic fungi [16,17,19,21,24,25,59]. The function of the COMPASS complex in mediating H3K4 methylation is highly conserved from yeast to humans; however, variations exist among the subunits of this complex. For instance, only a single Set1 protein exists in *S. cerevisiae*, while three are found in *Drosophila*, and six Set1-like methyltransferases are found in humans [7]. 

*C. neoformans*, the leading cause of fungal meningitis, is the top-ranked human fungal pathogen on the WHO Fungal Pathogen Priority List. Here, we conducted a systematic identification of COMPASS subunits and biological function analyses of these subunits in *C. neoformans* and *C. deneoformans*. We identified six conserved cryptococcal subunits that interact with each other and form the catalytic core of the COMPASS complex. Results from biochemical and genetic analyses demonstrate that the cryptococcal COMPASS complex governs H3K4me1, H3K4me2, and H3K4me3 modifications. In this particular pathogen, the COMPASS complex regulates thermal tolerance and melanin production, as well as virulence in the *G. mellonella* and murine models of cryptococcosis. It also regulates the yeast-to-hypha transition. Furthermore, COMPASS-mediated methylation of H3K4 is dependent on H2Bub1, which requires the PAF1C and Rad6/Bre1 ubiquitinating system. 

Regarding the regulation of the cryptococcal yeast-to-hypha transition, our results demonstrate that the COMPASS complex affects the transcription of genes that are critical for unisexual filamentation and bisexual mating, such as the master regulator of filamentation *ZNF2*, as well as genes encoding pheromones and pheromone receptors. The identification of COMPASS subunits thus provides new insights into mechanisms of epigenetic regulation of morphogenesis and pathogenesis in the human fungal pathogen *C. deneoformans*.

Our biochemical studies showed that common subunits shared by multiple species from yeast to humans, including Set1, Bre2, Spp101, Swd1, Swd2, and Swd3, form the catalytic core of the COMPASS complex in *C. deneoformans* (Figure 1 and Figure 2). Those subunits newly identified in *C. deneoformans* are especially highly conserved in fungi, though there are some important functional differences. In yeast, the ScSwd2 subunit has been reported to be essential [8,9], while it was found to be dispensable in *C. deneoformans*. This lack of essentiality is consistent with the findings of studies involving COMPASS in humans and other fungi, in which the *SWD2* homolog can be deleted. In *M. oryzae*, the deletion of *SWD2* results in a dramatic decrease in H3K4me3 and a deficiency in fungal development [16]. In humans, deletion of the WDR82, which is a homolog of ScSwd2, has been demonstrated to decrease H3K4me3 near the transcription start site of transcribed genes, and this deletion is associated with tumor progression [68,69]. These observations suggest that the Swd2 homologs may have different functions in different organisms. Another difference among fungi involves the yeast COMPASS complex subunit Shg1 [70]. We were unable to identify its homolog in *C. deneoformans* based on sequence similarity or domain conservation.

We were also unable to identify a homolog of ScSdc1 in *C. neoformans* and *C. deneoformans*. This result was surprising, given the reported conservation of this protein in humans and several model fungi. In yeast, two copies of ScSdc1 form a sub-complex with Bre2 [6], and loss of ScSdc1 leads to decreased H3K4 methylation by Set1 [71]. The conserved function of ScSdc1 also has been reported in mammalian cells, where the ScSdc1 homolog DPY30 has been shown to interact with ASH2L, which is a homolog of yeast ScBre2, in the regulation of H3K4 methylation [72]. Our phylogenetic analyses clearly showed the lack of an ScSdc1 homolog in Basidiomycota but not in Ascomycota (Figure 2D), suggesting a potential evolutionary loss of Sdc1 in Basidiomycota. Given the existence of an Sdc1 homolog in humans, it is also possible that the function of Sdc1 may be performed by other components of the COMPASS complex in *C. deneoformans*. Our 3D structural modeling of the catalytic core of COMPASS indicates that amino acids near the C-terminus of Bre2 form a prolonged helix, which might be able to function in stabilizing the core complex (Figure 2A). Further investigation is required to unveil the assembling mechanisms leading to the assembly of the COMPASS complex in *C. deneoformans*.

H3K4 methylation mediated by the COMPASS complex requires H2Bub1, which is catalyzed by Rad6-Bre1. Meanwhile, the PAF1C directly stimulates the deposition of H2Bub1 through an interaction between its subunit Rtf1 and Rad6 [60]. It has been reported that H2Bub1 induces conformational changes in the COMPASS complex, during which the physical association of Spp1 with the catalytic core is sufficient to support the H3K4 methylation activity [12]. Our findings showed that the PAF1C subunit Rtf1 and Rad6 mediate H2Bub1 and subsequent H3K4 methylation (Figure 5B). Furthermore, we provided evidence that supports the possibility that this histone crosstalk possibly regulates the transcription of genes that are critical for cryptococcal mating and yeast-to-hypha transition (Figure 6). It is also worth mentioning that transcript levels of two genes, *PFT1* (*CNE03110*) and *RAM1* (*CNF02370*) encoding proteins involved in the farnesyltransferase complex were positively regulated by COMPASS-mediated H3K4 methylation (Figure 6C). It has been published that the protein farnesyltransferase complex is required for the maturation of cryptococcal pheromone and mating hyphal formation [73,74]. In addition, it is known that the yeast-to-hypha transition is an adaptive response to internal and external signals. It will be of great interest to investigate the signal transduction pathways that modulate histone crosstalk and regulation of gene transcription in response to specific stimuli.

In conclusion, our biochemical, genetic, and proteomic analyses identified the conserved subunits of the COMPASS complex and functionally characterized their critical roles in mediating H3K4 methylation and the consequent regulation of cryptococcal gene expression and fungal development and virulence traits. This work fills a knowledge gap regarding COMPASS-mediated epigenetic regulation in a human fungal pathogen.

## Figures and Tables

**Figure 1 jof-09-00672-f001:**
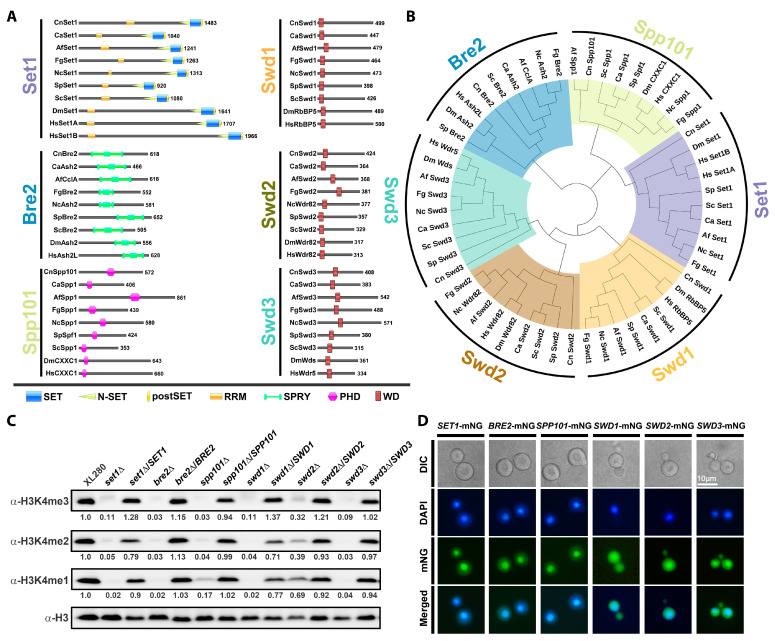
Phylogenetically conserved COMPASS subunits mediate H3K4 methylation in *C. deneoformans*. (**A**) A diagram depicting the domain structures of COMPASS subunits in selected fungi, fruit flies, and humans. Cn, *Cryptococcus neoformans*; Ca, *Candida albicans*; Af, *Aspergillus fumigatus*; Fg, *Fusarium graminearum*; Nc, *Neurospora crassa*; Sp, *Schizosaccharomyces pombe*; Sc, *Saccharomyces cerevisiae*; Dm, *Drosophila melanogaster*; Hs, *Homo sapiens*. The domain prediction was done with the InterPro option under the Protein Features menu in FungiDB. (**B**) Neighbor-joining tree of homologs of COMPASS subunits in selected fungi, fruit flies, and humans, generated based on their predicted protein sequences. Phylogenetic analyses were conducted with the whole-protein sequences with MEGA 7. (**C**) Western blotting analyses of histone H3K4me1, H3K4me2, and H3K4me3 in the wild-type XL280 strain and COMPASS subunit mutants. The pixel density of protein bands was measured by ImageJ as a gray level. The ratios of anti-H3K4 methylation band intensity over the corresponding band intensity of loading control (anti-H3) were calculated. The ratios of XL280 control were set as 1.0, and the ratios of mutants were normalized accordingly to determine the relative band intensity. (**D**) Fluorescence analyses of the sub-cellular localization of COMPASS subunits expressing an mNG-tagged version of the corresponding protein in its mutant background.

**Figure 2 jof-09-00672-f002:**
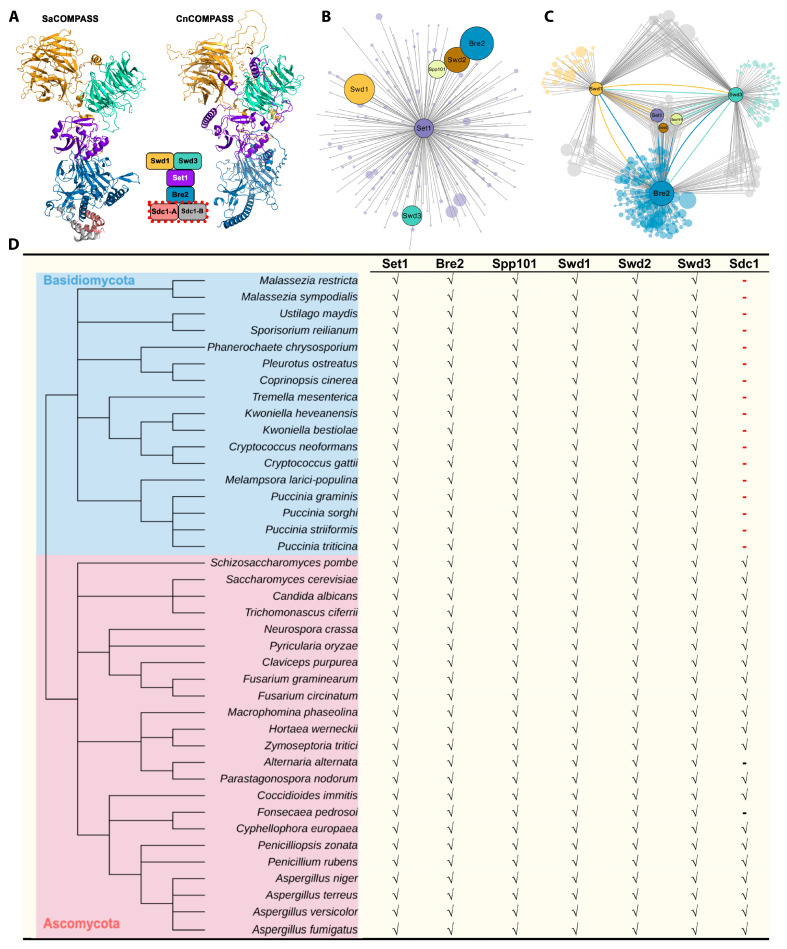
Set1, Bre2, Spp101, Swd1, Swd2, and Swd3 interact with each other and possibly form the integrated COMPASS complex in *C. deneoformans*. (**A**) 3D structural modeling of the catalytic core of COMPASS in *C. deneoformans* by AlphaFold-Multimer [59]. The yeast COMPASS histone methyltransferase catalytic module (PDB ID: 6chg) consisting of Swd1, Swd3, Bre2, Sdc1, and Set1 served as a comparison [6]. (**B**) Potential proteins that interact with Set1. A pull-down assay was performed with a strain expressing a FLAG-tagged version of Set1, and potential interacting proteins were identified by MS and visualized with Cytoscape software [60]. (**C**) Protein-protein interacting network of COMPASS subunits. FLAG-tagged versions of Bre2, Swd1, and Swd3 were used to perform a pull-down assay coupled with MS, respectively. The interacting network was visualized with Cytoscape software [60]. (**D**) A phylogenetic tree of selected fungi, including both ascomycetes and basidiomycetes. The phylogeny was conducted with the Taxonomy Common Tree tool based on the classification of the NCBI Taxonomy Database [61] and visualized with iTol [62]. The existence of COMPASS subunits for each fungus was indicated on the right. √ indicates the existence of a corresponding subunit; - in red or black indicates not identified based on sequence similarity.

**Figure 3 jof-09-00672-f003:**
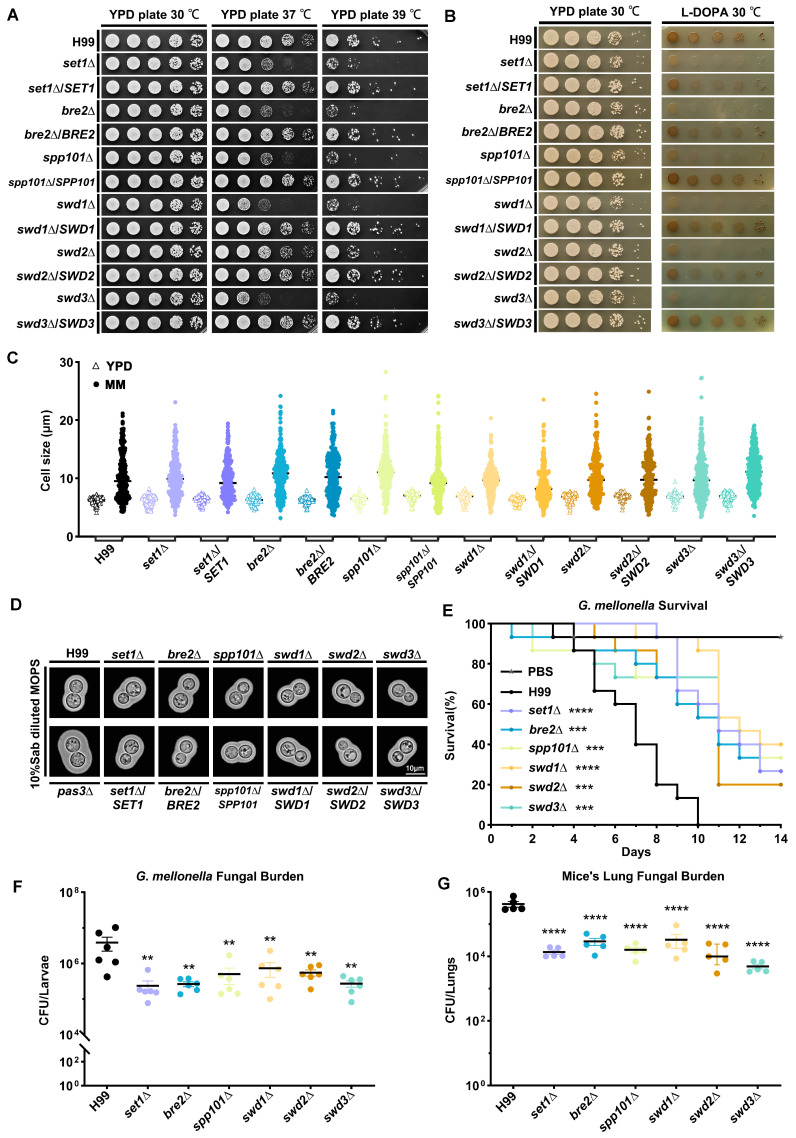
COMPASS subunits regulate virulence traits in *C. neoformans*. (**A**) Thermal tolerance spotting assays with wild-type H99 strain and COMPASS subunit mutant strains under indicated conditions. (**B**) Melanin production spotting assays with wild-type H99 strain and COMPASS subunit mutant strains on L-DOPA media. (**C**) Titan cell formation assay following the previously published protocol [52]. Cell size was measured using pictures taken in bright field and measured with ImageJ software. Given the relatively uniform cell size in YPD cultures, the size of 60 cells was measured as control, and 300 cells from titan cell-inducing MM condition (MM, 15 mM D-glucose, 10 mM MgSO4, 29.4 mM KH2PO4, 13 mM Glycine, 3.0 μM Thiamine, pH5.5) were measured. Each dot represents a single cell. (**D**) Indian ink staining of wild-type H99 strain and COMPASS subunit mutant cells grown on the capsule-inducing media. (**E**) The survival rate of infected *G. mellonella*. The survival curves were generated with Kaplan–Meier analyses, and the Gehan–Breslow–Wilcoxon test was used for statistical analyses of the survival data between mutant groups and H99 control. ****, *p*-values < 0.0001; ***, *p*-values < 0.001. (**F**) Fungal burden in *G. mellonella* larvae on day 5 post-infection. *G. mellonella* larvae were infected with COMPASS subunit mutant strains, the wild-type H99 strain, and PBS control. The one-way ANOVA tests were used in the fungal burden analyses. (**G**) The lung fungal burden of mice infected with H99 and COMPASS subunit mutants on day 14 post intranasal infection with 1 × 10^5^ cells/animal. The one-way ANOVA tests were used in the fungal burden analyses. ****, *p*-values < 0.0001; **, *p*-values < 0.01.

**Figure 4 jof-09-00672-f004:**
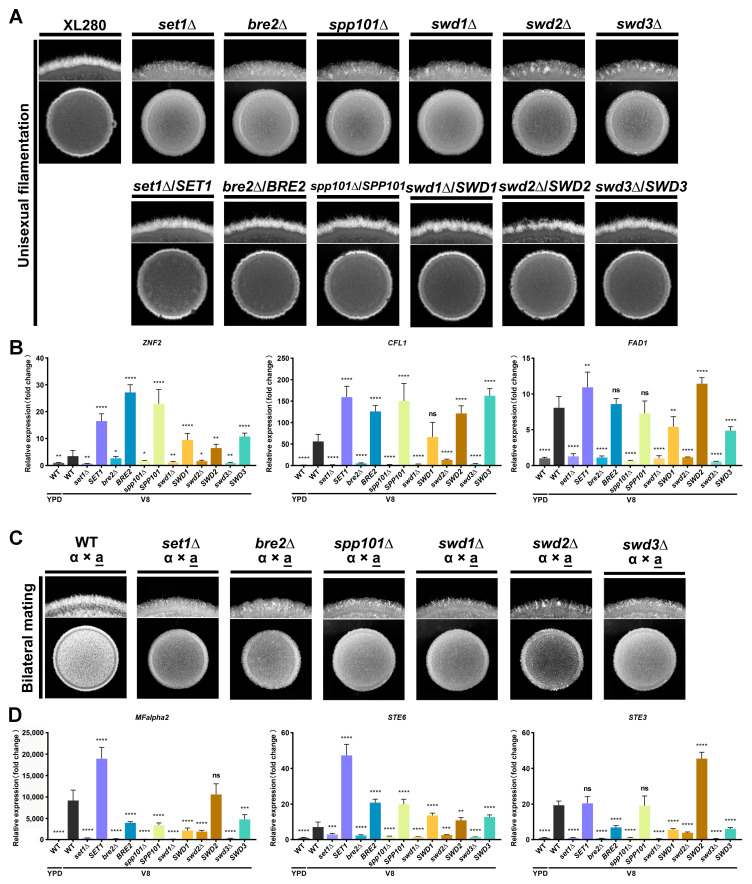
COMPASS subunits regulate the yeast-to-hypha transition in *C. deneoformans*. (**A**) Colony morphology of COMPASS subunit mutant strains and the wild-type XL280 strain on V8 media. The overnight cultures cells were washed and resuspended into OD_600_ = 3, and 3 μL of the resuspension for each strain was spotted onto V8 to monitor the yeast-to-hyphal transition. (**B**) Transcript levels of *ZNF2*, *CFL1*, and *FAD1* in cells from COMPASS subunit mutant strains and the wild-type XL280 strain cultured on V8 for 24 h. The transcript level of the corresponding gene from overnight cultures in liquid YPD served as a control for normalization and relative comparison. Three biological replicates were included in all assays. Student’s *t*-test was used for statistical analyses of corresponding gene transcript levels of mutants relative to that of WT under the V8 condition (the black bar). **** *p*-values < 0.0001; ** *p*-values < 0.01; * *p*-values < 0.05; ns *p*-values > 0.05. (**C**) Colony morphology of COMPASS subunit mutants and the wild-type XL280 strain during bilateral mating on V8 media. The overnight cultures of *MAT*α and *MAT***a** cells were washed and resuspended into OD_600_ = 3. The same volume of resuspensions indicating *MAT*α and *MAT***a** cells were mixed, and 3 μL of the mixture for each cross was spotted onto V8 to monitor the yeast-to-hyphal transition. (**D**) Transcript levels of *MFα2*, *STE6*, and *STE3α* in cells from COMPASS subunit mutant strains and the wild-type XL280 strain cultured on V8 for 24 h. The transcript level of corresponding genes from overnight cultures in liquid YPD served as a control for normalization. Three biological replicates were included in all assays. Student’s *t*-test was used for statistical analyses of corresponding gene transcript levels of mutants relative to that of WT under the V8 condition (the black bar). **** *p*-values < 0.0001; *** *p*-values < 0.001; ** *p*-values < 0.01; ns *p*-values > 0.05.

**Figure 5 jof-09-00672-f005:**
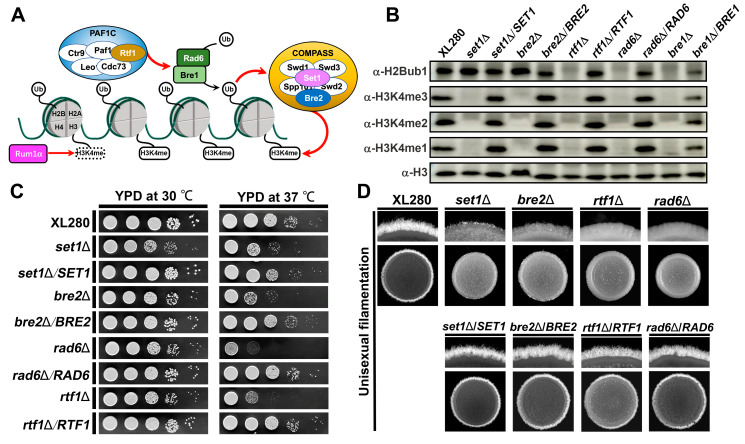
COMPASS-mediated H3K4 methylation requires PAF1C-mediated H2Bub1 in *C. deneoformans*. (**A**) A diagram depicting the regulatory crosstalk between COMPASS-mediated H3K4 methylation and PAF1C-mediated H2Bub1. Rad6 and Bre1 execute the mono-ubiquitination of H2B in a PAF1C-dependent manner. The catalytic core of the COMPASS complex executes the H3K4 methylation in an H2Bub1-dependent manner. (**B**) Western blotting analyses of H2Bub1 and H3K4 methylation in cells of *RAD6* and COMPASS and PAF1C subunit mutants. (**C**) Thermal tolerance assays of *RAD6* and COMPASS and PAF1C subunit mutant strains. The overnight cultures of cryptococcal cells were washed and resuspended into OD_600_ = 3. Ten times serial dilutions of the resuspensions were prepared, and 3 μL of the serial dilutions for each strain were spotted onto YPD media and cultured at specified temperatures for two days to monitor growth. (**D**) Colony morphology of *RAD6* and COMPASS and PAF1C subunit mutant strains on V8 media. The overnight cultures cells were washed and resuspended into OD_600_ = 3, and 3 μL of the resuspension for each strain was spotted onto V8 to monitor the yeast-to-hyphal transition.

**Figure 6 jof-09-00672-f006:**
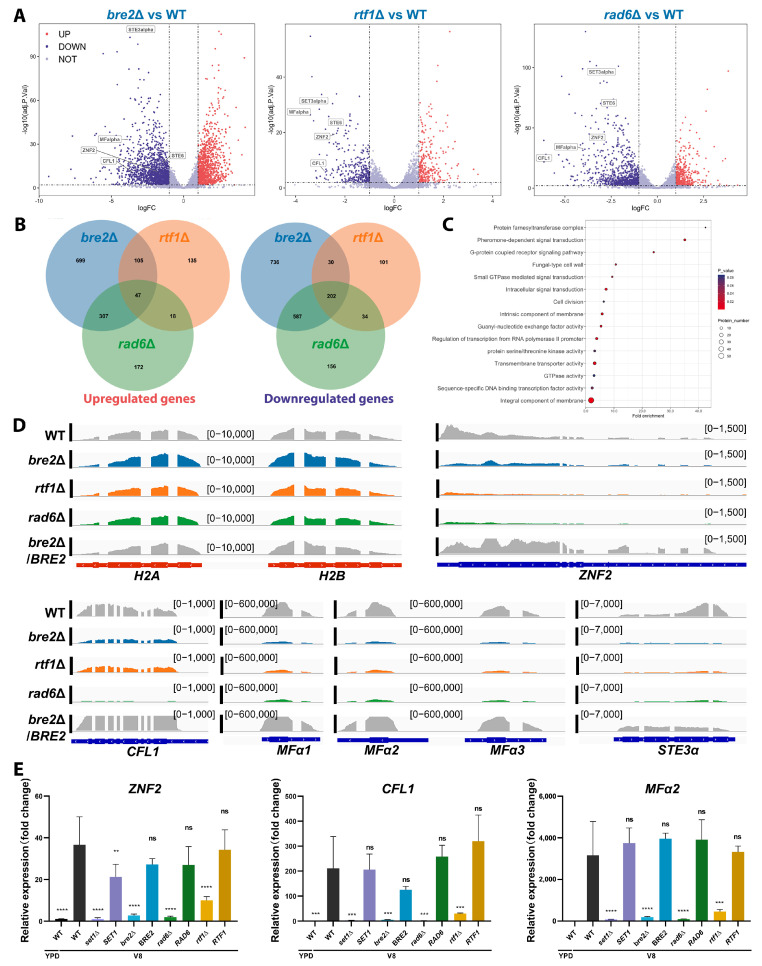
COMPASS subunit Bre2, PAF1C subunit Rtf1, and Rad6 share downstream regulons to promote filamentation in *C. deneoformans*. (**A**) Volcano plots of differentially expressed genes in mutants lacking *BRE2*, *RTF1*, or *RAD6* gene relative to the wild-type strain cultured on V8 media for 24 h. Each dot in the plots indicates a protein-coding gene. The vertical dash lines indicate the |log2FC| = 1, and the horizontal dash line shows the adjusted *p*-value = 0.05. The differentially up-regulated genes are indicated in red, and the differentially downregulated genes are indicated in blue. Genes related to mating and filamentation were labeled in the panels. (**B**) Venn diagrams of upregulated and downregulated genes in *bre2*Δ, *rtf1*Δ, and *rad6*Δ mutants. (**C**) The bubble plot indicates the significantly enriched GO categories of downregulated genes shared by *bre2*Δ, *rtf1*Δ, and *rad6*Δ mutants. (**D**) Reads coverage of indicated gene loci in *bre2*Δ, *rtf1*Δ, and *rad6*Δ mutants. Reads coverage at H2A-H2B loci in all strains and the complementation strain of *bre2*Δ served as controls. (**E**) qPCR quantification of transcript levels of *ZNF2*, *CFL1*, and *MFα*2 in *bre2*Δ, *rtf1*Δ, and *rad6*Δ mutants. Student’s *t*-test was used for statistical analyses of corresponding gene transcript levels of mutants relative to that of WT under the V8 condition (the black bar). ****, *p*-values < 0.0001; ***, *p*-values < 0.001; ** *p*-values < 0.01; ns, *p*-values > 0.05.

## Data Availability

The data that support the findings of this study are available on request from the corresponding author, Y.Z.

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
