# Peer review of "The COMPASS Complex Regulates Fungal Development and Virulence through Histone Crosstalk in the Fungal Pathogen Cryptococcus neoformans"

_jof, 2023, doi:10.3390/jof9060672_

Round 1
Reviewer 1 Report
In this study, entitled “The COMPASS complex regulates fungal development and virulence through histone crosstalk in the fungal pathogen Cryptococcus neoformans” It has successfully been identified the fundamental subunits comprising the COMPASS complex in both C. neoformans and C. deneoformans, in addition to their conserved functions in the methylation of H3K4, using a combination of AlphaFold modeling, molecular genetics, and biochemical approaches. This research provides evidence of the epigenetic regulation of cryptococcal development and virulence by the COMPASS complex, which involves histone crosstalk between H2Bub1 and H3K4 methylation.
Authors select fungi species of C. neoformans and C. deneoformans because they are the primary causative agents of fungal meningitis, holds the highest position among human fungal pathogens on the WHO Fungal Pathogen Priority List. They successfully identified six conserved subunits in cryptococcal organisms that not only interact with one another but also collectively constitute the catalytic core of the COMPASS complex. By identifying the subunits of the COMPASS complex, this study gained valuable insights into the mechanisms underlying the epigenetic regulation of morphogenesis and pathogenesis in the human fungal pathogen C. deneoformans. This discovery opens up new avenues for understanding the intricate processes involved in its development and disease-causing capabilities. Introduction is well written and described the main aim of the work. Methodology and experimental design sections were correctly designed and arranged. Discussion section has been written in an interesting way. There are interested issues in the current study which led me to judge the present manuscript as “accepted”.
1. What is the main question addressed by the research?
The main questions addressed by the research are: What are the core subunits of COMPASS complex in C. neoformans and C. deneoformans (as human fungal pathogen)?
Are these core subunits interacting with each other and function in regulating cryptococcal morphogenesis and virulence traits?
2. Do you consider the topic original or relevant in the field? Does it address a specific gap in the field?
It is original topic and address a specific gap in the field. The role of the COMPASS complex in regulating gene expression through histone methylation has been extensively studied and established as crucial in numerous biological processes such as it exerts regulatory control over fungal morphogenesis, stress responses, and virulence.
While the COMPASS complex has been extensively studied in different fungi such as model yeasts, plant fungal pathogen Magnaporthe oryzae, Alternaria alternata, Fusarium graminearum, Aspergillus, Colletotrichum higginsianum, and human fungal pathogen Candida spp. Its exploration in human fungal pathogens remains limited and requires further investigation.
“ Yang C, Huang Y, Zhou Y, Zang X, Deng H, Liu Y, Shen D, Xue X. 2022. Cryptococcus escapes host immunity: What do we know? Front Cell Infect Microbiol 12:1041036
So YS, Yang DH, Jung KW, Huh WK, Bahn YS. 2017. Molecular Characterization of Adenylyl Cyclase Complex Proteins Using Versatile Protein-Tagging Plasmid Systems in Cryptococcus neoformans. J Microbiol Biotechnol 27:357-364.
Trevijano-Contador N, de Oliveira HC, García-Rodas R, Rossi SA, Llorente I, Zaballos Á , Janbon G, Ariño J, Zaragoza Ó . 2018. Cryptococcus neoformans can form titan-like cells in vitro in response to multiple signals. PLOS Pathogens 14:e1007007
Trevijano-Contador N, de Oliveira HC, García-Rodas R, Rossi SA, Llorente I, Zaballos Á , Janbon G, Ariño J, Zaragoza Ó . 2018. Cryptococcus neoformans can form titan-like cells in vitro in response to multiple signals. PLOS Pathogens 14:e1007007
Lin J, Fan Y, Lin X. 2020. Transformation of Cryptococcus neoformans by electroporation using a transient CRISPR-Cas9 expression (TRACE) system. Fungal Genet Biol 138:103364.
Stempinski PR, Smith DFQ, Casadevall A. 2022. Cryptococcus neoformans Virulence Assay Using a Galleria mellonella Larvae Model System. Bio Protoc 12 “
3. What does it add to the subject area compared with other published material?
While the COMPASS complex has been extensively studied in different fungi such as model yeasts, plant fungal pathogen Magnaporthe oryzae, Alternaria alternata, Fusarium graminearum, Aspergillus, Colletotrichum higginsianum, and human fungal pathogen Candida spp. Its exploration in human fungal pathogens remains limited and requires further investigation.
C. neoformans and C. deneoformans were chosen as the fungi species of focus by the authors due to their status as the primary causative agents of fungal meningitis. These fungi hold the highest rank amongst human fungal pathogens on the WHO List.
4. What specific improvements should the authors consider regarding the methodology? What further controls should be considered?
The methodology and experimental design sections were correctly designed and appropriately structured to ensure accuracy and reliability in the study. Appropriate controls are considered and included in the methodology and results sections; for example, in figures 3, 4, 5, and 6.
5. Are the conclusions consistent with the evidence and arguments presented and do they address the main question posed?
Yes. The conclusion of this study is consistent with the evidence and arguments presented, and it specifically addresses the main question posed.
6. Are the references appropriate? Yes
7. Please include any additional comments on the tables and figures.
While the main body of the work does not include tables and the figures can be complicated, the authors have thoughtfully included numerous supplementary tables and figures in the supplementary data. These additional resources provide detailed information that enhances the explanations presented in the figures.Author Response
Dear reviewer, First of all, thank you for taking the time to read and modify this article. Thank you for your valuable suggestions. I carefully studied your opinions and carefully revised the paper according to the suggestions.Reviewer 2 Report
I suggest that the authors review the text, and mainly, the figures abbreviations to clearly differentiate the data with C. deneoformans from the ones with C. neoformans. This is most times confusing and compromise the fluency of the reading and the understanding of the results.
Author Response
Dear reviewer, First of all, thank you for taking the time to read and modify this article. Thank you for your valuable suggestions. I have carefully took into your opinions and carefully revised the paper according to the recommendations, as follows: We have revised the text, and especially, the figures abbreviations to clearly differentiate the date with C. deneoformans from the ones with C. neoformans.Reviewer 3 Report
The manuscript of Liu and colleagues aimed to identify the subunits of the COMPASS complex in two Cryptococcus spp. and their role in mediating histone H3K4 methylation. The results have shown the involvement of these subunits in regulation of virulence traits, such as, morphogenesis, temperature tolerance and melanin production.
The manuscript is well written and very interesting. They used well-described phenotypic, genotypic and proteomic approaches that were essential to obtain the main conclusions. I have only some comments that I would like to be addressed:
- Why did the authors use two animal models by using G. mellonella and Balb/c mice? In my opinion G. mellonella model is redundant.
- In the murine model the authors perform intranasal inoculation and perform lung fungal burden. Wouldn`t it be more interesting to brain fungal burden resulting from a systemic model of infection, since Cryptococcus neoformans is an opportunistic fungal pathogen responsible for meningoencephalitis? Please comment.
- Since COMPASS subunits regulate yeast-to hypha transition, their involvement in adherence and biofilm formation should be studied.
- No information regarding the antifungal susceptibility profile of the isolates is shown. The authors should comment on the involvement of COMPASS subunits in antifungal susceptibility and oxidative stress response.
Author Response
Dear reviewer, First of all, thank you for taking the time to read and modify this article. Thank you for your valuable suggestions. I carefully studied your opinions and carefully revised the paper according to the suggestions.The responses are as follows, To question #1: The G. mellonella and Balb/c mice are different animal models, and their infection results confirm each other. To question #2: The strain of Cryptococcus neoformans used in the paper are weaker for brain infections than lungs. To question #3: We can further explore COMPASS subunits' involvement in adherence and biofilm formation. To question #4: At present, we have no experimental results on the antifungal drug sensitivity of the isolates, and this content can be explored as a direction in the future.Reviewer 4 Report
In the manuscript by Liu R. et al., the authors identify and characterize the Complex of Proteins Associated with Set1 (COMPASS) in C. neoformans. This is the first study that identified all the major components of COMPASS in this species. The authors made deletions of 6 identified genes predicted to encode proteins that are part of the COMPASS complex. Consistent with being COMPASS components, these proteins were necessary for methylations of histone H3 on lysin 4 and localized to nucleus. As previously shown in other fungal models, identified genes were necessary for stress response, including growth at host temperature and filamentation during mating (also unisexual mating). The authors provide evidence the filamentation defect is due to inability to regulate pheromone sensing pathway. COMPASS genes are also shown to be important for virulence based on Galleria and murine infection models and deduced from defects of the mutants in producing melanin. In contrast, the authors conclude that COMPASS is not necessary for formation of Titan cells and capsule formation. The authors also use mass spectrometry to identify proteins that associate with the identified COMPASS components. Interestingly, the authors provide several lines of evidence that COMPASS component Scd1, present in ascomycetes, is absent in C. neoformans and apparently also other basidiomycetes. Finally, the authors demonstrate that, like in other previously described species, COMPASS-mediated H3K4 methylation requires H2B monoubiquitination in C. neoformans. Overall, this is a very comprehensive, and generally well-written and well-presented study. Here are my comments that may help to improve this manuscript:
1. I suggest, the Authors do a thorough grammar/spell check as I found occasional errors that need correction.
2. I found that the reference in figure legend to Figure 3C is incorrect (currently it is 40 and it should be one of the 50-52).
3. In Figure 3C legend – what is the “MM condition”?
4. The description of capsule protocol in Materials and Methods (Ln 207) is misleading as it suggests that the cells were spotted on a semi-solid capsule inducing medium. I suspect liquid capsule inducing medium has been used, just like in ref #50.
5. I suggest the Authors double check that no unnecessary references are included in the manuscript – for instance, not all three #60-62 references are necessary to illustrate the importance of Titan cells, especially that the mutants had no effect on Titan cell production.
6. I found no figure legends to supplementary figures. For instance, it needs to be specified which specific method has been used to generate data for each of the following: Figure 3C, and S4A and S4B.
The text needs additional editing as some errors were found
Author Response
Dear reviewer, First of all, thank you for taking the time to read and modify this article. Thank you for your valuable suggestions. I carefully studied your opinions and carefully revised the paper according to the suggestions.The responses are as follows, To question #1: We have done a thorough grammar/spelling check and corrected some mistakes. To question #2: We have corrected the reference to Figure 3C in the legend to [52]. To question #3: We have added the expression of “MM condition” in Fig 3C. To question #4: We have revised the description of capsule protocol in Materials and Methods (Ln 207). To question #5: Because the current in vitro induction methods of Titan cells vary greatly in different laboratories and under different conditions, we chose the three methods involved in the cited literature to determine the induction of mutant Titan cells. To question #6: We have added figure legends to supplementary figures.